# Polar cod in jeopardy under the retreating Arctic sea ice

Mats Brockstedt Olsen Huserbråten [1]*, Elena Eriksen[1], Harald Gjøsæter [1] & Frode Vikebø[1,2]

The Arctic amplification of global warming is causing the Arctic-Atlantic ice edge to retreat at unprecedented rates. Here we show how variability and change in sea ice cover in the Barents Sea, the largest shelf sea of the Arctic, affect the population dynamics of a keystone species of the ice-associated food web, the polar cod (*Boreogadus saida*). The data-driven biophysical model of polar cod early life stages assembled here predicts a strong mechanistic link between survival and variation in ice cover and temperature, suggesting imminent recruitment collapse should the observed ice-reduction and heating continue. Backtracking of drifting eggs and larvae from observations also demonstrates a northward retreat of one of two clearly defined spawning assemblages, possibly in response to warming. With annual to decadal ice-predictions under development the mechanistic physical-biological links presented here represent a powerful tool for making long-term predictions for the propagation of polar cod stocks.

[1] Institute of Marine Research, Box 1870 , 5817 Bergen, Norway. [2] Bjerknes Centre of Climate Research, Box 7810 , 5020 Bergen, Norway. *email: mats.huserbraaten@imr.no

The Arctic Ocean winter sea ice cover has steadily declined since the 1970s, and the majority of this reduction has been observed in the Barents Sea[1–4]. With a warmer and increasingly ice-free Arctic Ocean, the border between the Boreal and Arctic biomes is predicted to move north[5,6]. While the present warming has allowed an opportunistic northwards expansion of Boreal species into the northern Barents Sea Arctic ecosystem[7], little attention has been given to the effect of the recent ice retreat and variability on the ice-associated fish community. A keystone species in the ice-associated Arctic marine food web is the polar cod, one of few species linking the lower (i.e. zooplankton) and higher (e.g. other fish, mammals, seabirds) trophic levels[8–10]. The polar cod is endemic to the Arctic and their early life history is strongly adapted to the presence of ice: from spawning of eggs under the ice[11]; the ability of eggs to develop in sub-freezing temperatures[12]; larvae feeding on zooplankton specific to the seasonal ice-melt-water blooms[13,14]; and low mortality of larvae in the close to freezing temperatures typical of Arctic water masses[15]. At the same time observations made by Soviet-era researchers in the early 1960s[11] remain the most complete descriptions of polar cod spawning along the Eurasian shelf, with a historical stronghold in spawning activity in the south-eastern corner of the Barents Sea (also known as the Pechora Sea) and a suggested spawning east of Svalbard. The Barents Sea polar cod stock have been monitored annually by a joint Norwegian-Russian survey since 1986[16], and the total stock biomass (TSB) has varied vastly in the past three decades, between a minimum of 127,000 t in 1990, up to a maximum of 1,941,000 t in 2006 yet with no clear trend[17]. The spatial distribution and abundance of 0-group polar cod (~6 months old) has also varied considerably in this period[18], suggesting high recruitment variability both in space and time.

Given the tightly linked early life history of the polar cod with ice, a natural candidate for the large variation in polar cod recruitment and biomass is the inter-annual variation in ice cover[18]. The two main factors driving the observed variability in ice cover in the Barents Sea is the inflow of warm Atlantic water from the west and the inflow of cold, less saline Arctic water from the north and east[4,19,20]. The inflow of Atlantic water from the west and Arctic water from the northeast set up a strong temperature and salinity front across the entire Barents Sea, here termed the Polar Front[21,22] (Fig. 1). North of the Polar Front the less dense Arctic water masses stabilizes the water column sufficiently to prevent upwards flux of the warmer Atlantic water, allowing ice to form[4]. In March-April the extent of the seasonal ice cover in the Barents Sea is at its largest, and at this time period the Polar Front and the ice edge usually coincide. During 1990–2017 the winter maximum ice cover in the Barents Sea varied between 632,304 km$^2$ and 1,129,719 km$^2$, with an average of 869,961 km$^2$.

Here we investigate possible links between inter-annual variability in the Barents Sea ice cover and the variability in spawning distribution and recruitment of polar cod. To locate the spawning locations of polar cod under the ice we "back-tracked" larval drift trajectories from observed 0-group polar cod in autumn to the most probable spawning locations in spring. This

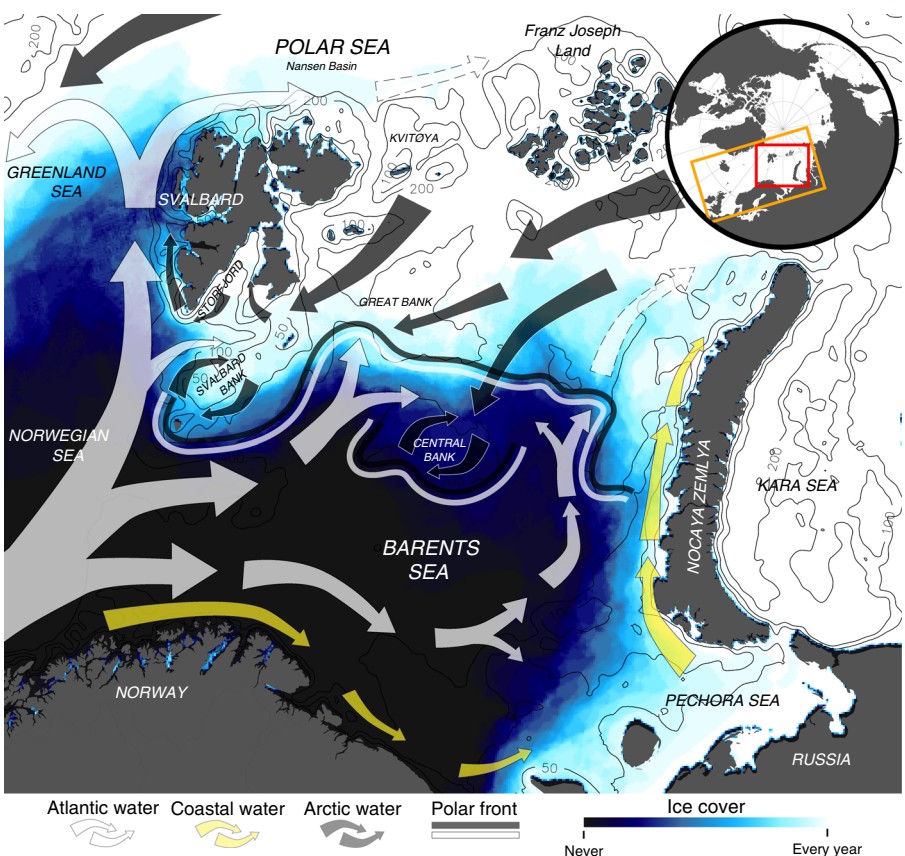

**Fig. 1** Oceanographic features of the study area and modelled maximum ice cover in the study period (1990–2017). Here white, yellow, and grey arrows represent the idealistic flow of near surface (0–10 m) Atlantic, coastal, and Arctic water masses (white arrows with perforated edges represents subsurface flow); double white and grey line represent the Polar Front; and the colored gradient represents proportion of years in the study period where ice concentration exceeds 15% per grid cell. Black lines represent isobaths at 50, 100, and 200 m. Orange and red squares in small inset represents the spatial extent of the ocean model and the study area, respectively

was done by Lagrangian particle advection simulations within a 3D dynamical ocean model. Drifting particles were "spawned" uniformly over the parts of the Barents Sea that was encompassed by the historical marginal ice zone (i.e. where ice concentration had been at least 15% at any time in period 1990–2017), and subsequently an objective search algorithm identified drift trajectories that intersected the 0-group observations of the autumn survey. The coupled biophysical model was simulated over 28 spawning seasons (1990–2017) and was compared to the catch of 0-group polar cod in 8302 pelagic trawls distributed throughout the Barents Sea. The mechanistic interpretation of the scores from the objective search algorithm is twofold: first the backtracking from observations indicates spawning at a given release point for a given year (viz. inter-annual change in spawning location); and second the frequency of links between spawning location and the observed 0-group abundance reflects larval survival during the drift phase and indicates supply of recruits from a given spawning location (viz. recruitment variability in response to environmental conditions). We report a positive relationship between year class strength of polar cod to the inter-annual variability of ice cover, and a negative effect of maximum summer temperatures encountered in late larval phase, where a common driver is the heating of the system. We also observed a clear northward retreat of one the two identified spawning assemblages towards the end of the study period that may also be a response to warming. Understanding the physical-biological interactions during recent decades of both varying and changing ice conditions allow predictions of future development of the polar cod stock in the Barents Sea.

## Results

**Inter-annual change in spawning location**. The objective search algorithm revealed two clearly separated maxima in spawning location probability, with one spawning assemblage in the Pechora Sea and one east of Svalbard. Moreover, the location of the two main spawning areas identified coincided with the high probability of 0-group occurrence downstream to the two areas (Generalized additive model (GAM) explaining 40% of the variation in 0-group presence/absence, Fig. 2). At the same time there was considerable inter-annual variability in the location of the center of the back-calculated main spawning areas. For example, in 1990 when the observed spatial distribution of 0-group polar cod was strongest correlated with spawning east of Svalbard, most of the eggs and larvae drifted with the East Spitsbergen Current into the Storfjord, further along the west coast of Svalbard, or onto the Svalbard Bank, where most of the 0-group polar cod was found that year (Fig. 3a, and see Fig. 1 for references to location). Recruitment in 1995 was very low, indicated by a record low abundance of 0-group polar cod found in the autumn cruise, coinciding with the Pechora Sea largely being ice-free during most of winter and spring (Fig. 3b) and maximum temperatures encountered in late summer exceeding 10 °C. This in contrast to 1998 when the ice cover in the south-eastern Barents Sea was at its highest in the study period, and the release

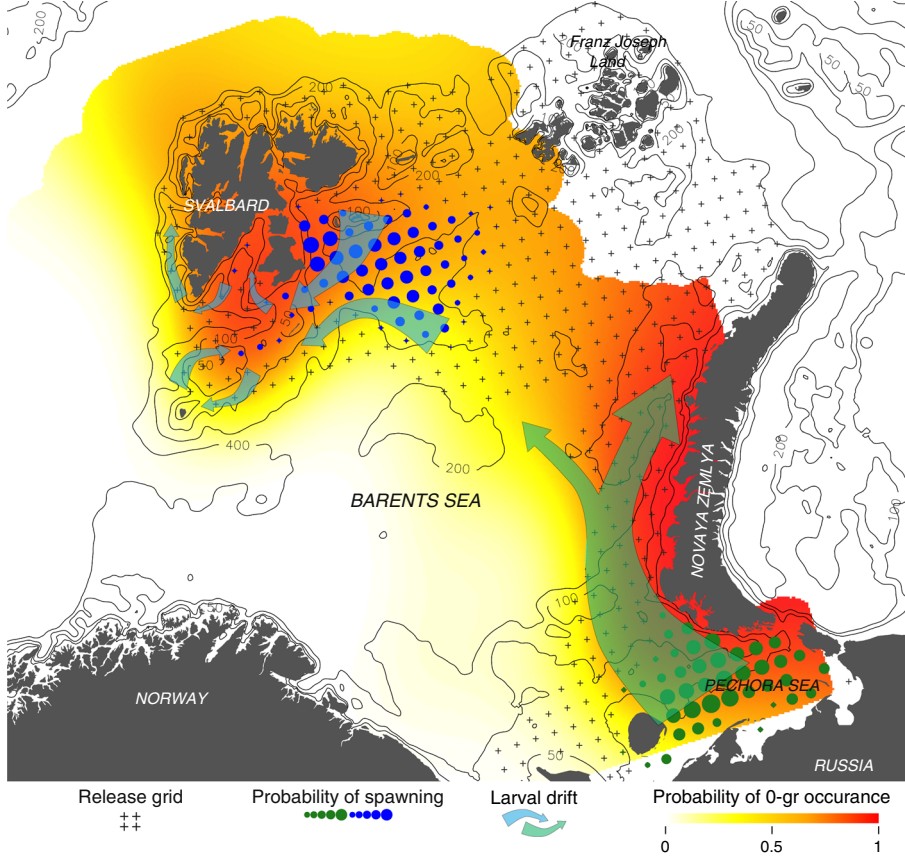

**Fig. 2** Predicted polar cod spawning, idealized drift routes, and probability of 0-group occurrence. Here black crosses represent the initial release points for the Lagrangian particle advection simulations; blue and green dots represent the integrated probability of spawning at a given release points across the study period; and arrows represents the idealized drift routes of eggs and larvae. Colored gradient represents the probability of 0-group occurrence in trawl hauls, predicted from the 2D GAM fitted to the geographical position of 0-group presence/absence explaining 40% of the deviance. Note that the sharp boundary of the colored gradient to the north and east reflects the extent of sampling rather than the actual distribution of 0-group polar cod. Black lines represent isobaths at 50, 100, 200, and 400 m

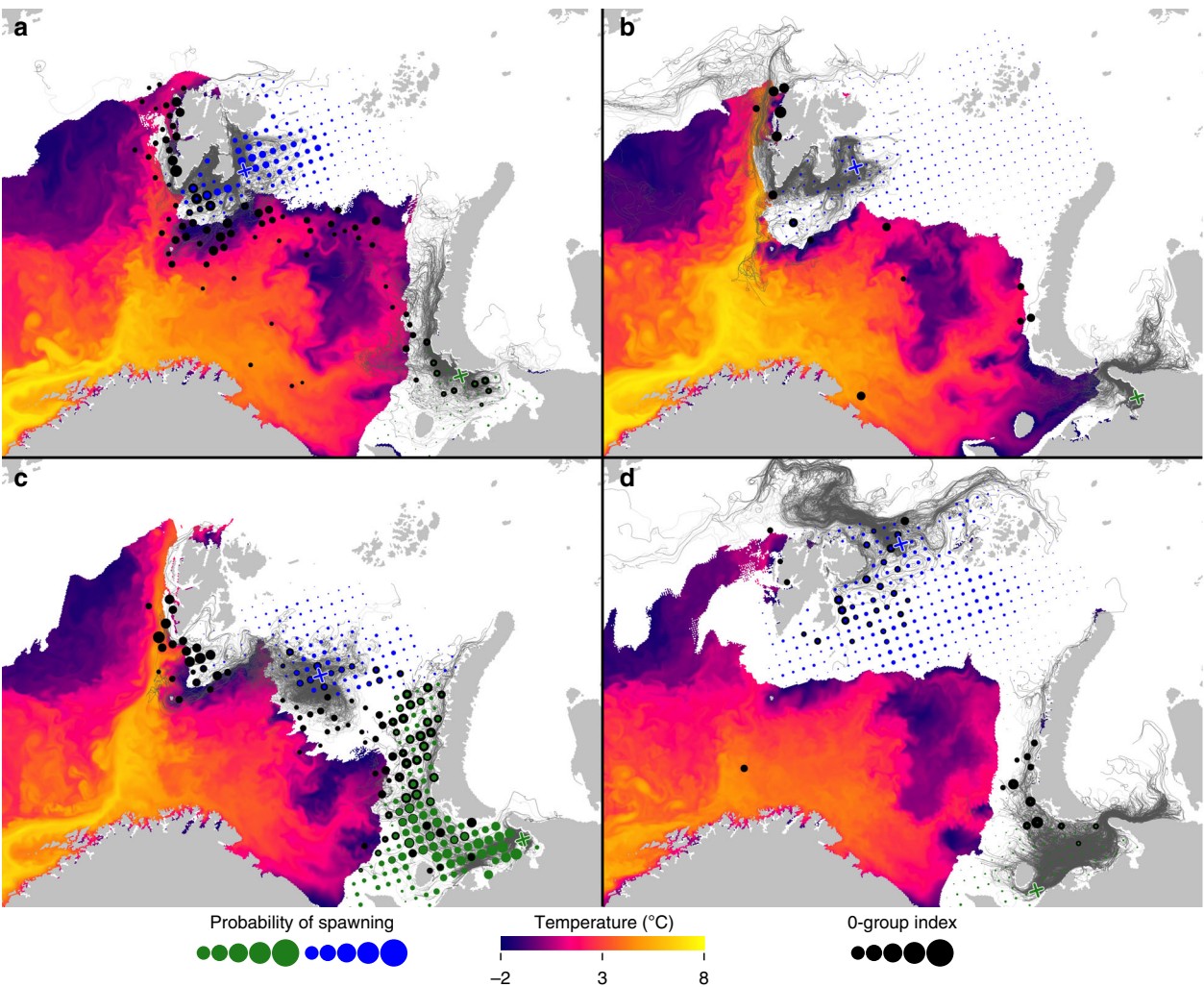

**Fig. 3** Drift trajectories of polar cod eggs and larvae from most likely spawning location in **a** 1990, **b** 1995, **c** 1998, and **d** 2013. Here size of green and blue dots represents the individual scores from the search algorithm of each release point, and black dots represents the observed (scaled) 0-group abundance. Grey lines represent 100 individual drift trajectories released from the location that had the best score for each year, here indicated by a blue/green cross for eastern Svalbard and Pechora Sea component, respectively. Also depicted is the modelled annual maximum marginal ice cover (white area, i.e. where ice concentration exceeds 15% per grid cell) and modeled sea surface temperature at the time of maximum ice cover (colored gradient)

points in the Pechora Sea was strongly correlated with the high abundance of 0-group polar cod found in a wide swath along Novaya Zemlya (Fig. 3c). Perhaps the most anomalous year in terms of drift patterns was in 2013 when the most probable spawning location of the eastern Svalbard spawning assemblage was situated north of Kvitøya, close to the shelf edge towards the Nansen Basin–yielding a qualitatively different dispersal pattern than previous years, where the majority of larvae initiated from this particular location were advected along the margins of the continental shelf instead of into the Barents Sea (Fig. 3d). This northern displacement of the eastern Svalbard spawning assemblage in 2013 was not an isolated instance, with a similar displacement also in 2009, 2015, and 2017. The predicted western spawning center was also located on the Great Bank in many of the years in the middle of the study period (2000s), and even near the Central Bank in some years. The average northwards displacement towards the end of the study period (2015–2017) constituted 2° of latitude (approx. 220 km) compared to the yearly 2000s (2001–2005), culminating a clear decadal northward trend (Fig. 4).

**Recruitment variability in response to ice loss and heating.**
Recruitment strength in the eastern Barents Sea/Pechora Sea had a significant correlation with maximum yearly ice cover (r = 0.69, t = −4.49, df = 26, p < 0.001, see Fig. 5), reiterating the importance of ice for early pelagic stages. The recruitment in the Pechora Sea was also significantly correlated with the estimated TSB two and a half years later (r = 0.61, t = 3.58, df = 22, p < 0.001), confirming the Pechora Sea spawning assemblage as the main supplier of recruits to the Barents Sea stock monitored during autumn surveys. Recruitment east of Svalbard had no correlation with either ice cover, temperature, or TSB at any lag, indicating other effects at play than those tested within the scope of this study. A linear regression of maximum ice cover over the study period showed a weak, yet significant yearly decrease of 1.1% (F = 4.93 on 1 and 26 df, p = 0.035, R$^2$ = 0.16). Moreover, the 10-day mean-filtered maximum temperature encountered by larvae in late summer was significantly different between the two main spawning areas (t = −7.46, df = 47, p < 0.001), with maximum temperatures encountered in the eastern Barents Sea on average between 2.1 °C and 3.6 °C degrees warmer (95% CI).

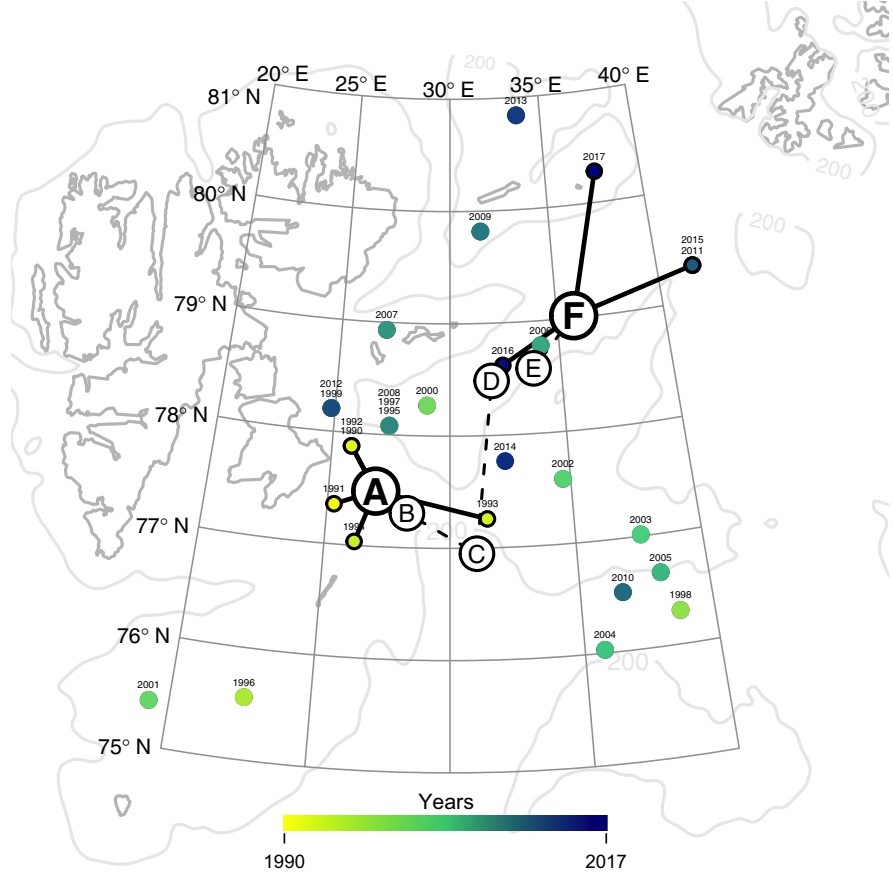

**Fig. 4** Northward displacement of spawning east of Svalbard. Here colored dots and gradient represents the position of the most likely spawning area each year of the study period, and letters A–F the five-year average position of most likely spawning location (A: 1990–1994, B: 1995–1999, C: 2000–2004, D: 2005–2009, E: 2010–2014, and F: 2015–2017). Plotted in grey is the contour of the Svalbard archipelago and the 200 m isobath

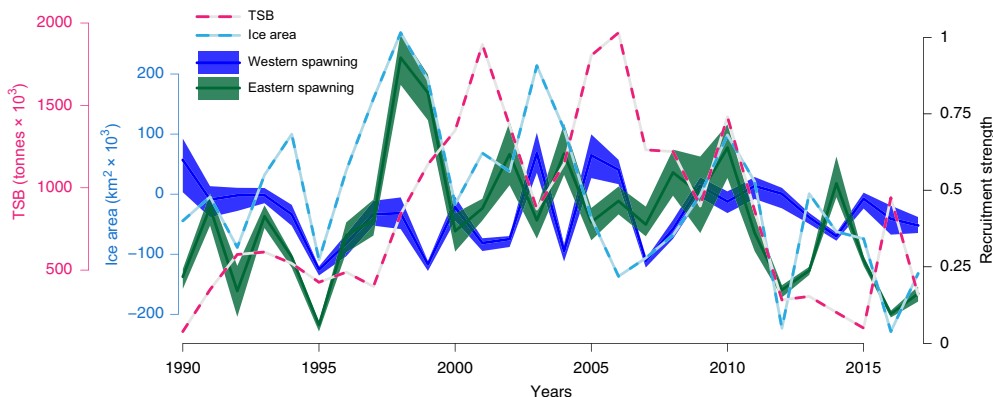

**Fig. 5** Total stock biomass (TSB), maximum ice cover, and the back-calculated recruitment strength from spawning assemblage east of Svalbard and Pechora Sea. Here estimated TSB is measured in thousand metric tonnes and represented by a pink line, and the light blue line represents the annual maximum marginal ice cover (i.e. area with ice concentration above 15% per grid cell)

The average yearly maximum temperature encountered in the Pechora/eastern Barents Sea was 5.7 °C but in extreme years more than 9 °C (95% CI: 2.4–9.0 °C), well beyond the thermal tolerance limit of the larvae (≈100% mortality at extended periods above 7 °C[15]). A linear regression model of eastern Barents Sea recruitment strength as function of covariates Barents Sea ice cover, maximum temperature encountered by larvae, and TSB, explained 72% of the variation in recruitment strength (F = 20.7 on 3 and 23 df, $p < 0.001$). Ice and temperature explained the majority of the variance, and a smaller portion explained by TSB ($R^2_{ice\ cover} = 48\%$, $R^2_{max\ temp} = 44\%$, $R^2_{TSB} = 16\%$). However, the two variables ice cover and temperature where found to share a high portion of the explained variance ($R^2_{ice\ cover} \cap R^2_{max\ temp} = 25\%$), albeit being significantly negatively correlated (r = −0.38, t = −2.09, df = 26, $p = 0.045$).

## Discussion

We suggest two plausible mechanisms driving the large recruitment variability observed in the south-eastern parts of the Barents Sea, mediated in turn by both heat and ice. First, the presence/absence of ice likely causes a "match/mis-match" scenario[23,24] specific to the seasonally ice-covered Arctic food web. The primary food item for the first-feeding larval phase is the naupliar stages of Arctic copepod *Calanus glacialis* that are nourished by the algae bloom confined to the meltwater plume[13,25,26]. A mismatch most likely arises in years with reduced or no ice cover in the Pechora Sea when the phenology of the spring bloom follows Atlantic-like bloom dynamics initiated by thermal stratification rather than ice-melt[25,27,28]. However, if larvae indeed survive that first critical ice-associated phase all three major *Calanus* species found in the Barents Sea are eaten indiscriminately[13]. This includes the highly abundant *C. finmarchicus* and the larger (but rarer) *C. hyperboreus*, both advected from the west with the warmer Atlantic water masses, and all three *Calanus* species having exceptionally high energy content[29]. The ample food supply available to the later larval stages in the Atlantic-influenced water masses found in the south-eastern parts of the Barents Sea thus represents a near ideal nursery habitat. At the same time, the high summer temperatures sometimes encountered there may also be associated with high risk by pushing larvae beyond thermal thresholds[15]. For example, the seven warmest years in terms of maximum summer temperatures encountered (as observed in 1990, 1995, 2000, 2012–2013, 2016–2017), resulted in the five years of lowest recruitment. However, due to the collinearity of ice cover and high temperatures encountered in south-eastern Barents Sea during drift in late summer, where a common driver is the wind-driven advection of warmer water masses from the west[20], we were not able to disentangle which of the two identified mechanisms that affected recruitment in the Barents Sea the most.

Regional downscaling of future climate change scenarios predicts an increased frequency of years with reduced or no ice cover in the south-eastern Barents Sea towards 2050[30,31]. However, due to the strong topographic steering of the incoming Atlantic water masses away from the northern parts of the Barents Sea (cf. Figure 1), the cold and icy winter-conditions currently found east of Svalbard is likely to persist into the foreseeable future[19,32,33]. The predicted deterioration of the south-eastern Barents Sea nursery habitat will thus increase the relative importance of the supply of recruits coming from the eastern Svalbard spawning assemblage in the future. At the same time the observed northward displacement of the eastern Svalbard spawning assemblage towards the end of the study period may qualitatively alter the dispersal pathway of eggs and larvae. The advection of the surface layer in the far northern Barents Sea area is mainly influenced by the prevalent wind field, where the meltwater and pack ice generally follow a net westward direction towards the Fram Strait[19,20]. Other potential dispersal pathways that may be realized in the northernmost Barents Sea include getting entrained with the Atlantic boundary current flowing along the Nansen Basin[34], or lateral transport towards the Arctic Ocean interior with eddies detaching from the boundary current itself[35]. In any case, polar cod larvae and juveniles drifting with the pack ice in the Arctic Ocean interior is not an uncommon phenomenon[36], although the fate of these drifting individuals is not known.

The biophysical links highlighted here represent fundamental knowledge of how a keystone species of the Arctic food web respond to contemporary climatic forcing. As the frequency of years with low or no ice cover in the south-eastern Barents Sea is predicted to increase towards the middle of the millennium, we expect recruitment of polar cod to the south-eastern Barents Sea to become even more variable or diminished, most likely having dramatic consequences for the entire food web. And ultimately if summer temperatures continue to exceed the thermal tolerance of larvae, the south-eastern Barents Sea as a nursery area will be unsuitable altogether. Conversely, the topographical steering of Atlantic water masses away from the north-western Barents Sea will most likely facilitate ice and spawning east of Svalbard even in a warmer climate than today.

## Methods

**Recruitment and spawning stock biomass indices**. The 0-group polar cod data were sampled on annual surveys run between late August and early October in the period 1990 and 2017, covering almost the entire Barents Sea within a regular grid of ~65 km. At each station the upper water layer (0–60 m) was sampled by three pelagic trawls with a 20 × 20 m opening, keeping the headlines at 0 m, 20 m, and 40 m. The pelagic trawls were towed at a speed of 3 knots over a time interval of 10 min, corresponding to a tow length of 0.5 nautical miles (≈0.93 km). If dense concentrations of fish appeared on the echo-sounder deeper than 40 m, additional tows were performed at 60 and 80 m. During the study period of 27 years 8302 of these depth-integrated trawl hauls were done. Due to the selectivity of the gear[37], the catches were adjusted for capture efficiency using a stratified sample mean method[38,39]. As a proxy for total stock biomass (TSB) we estimated the total mass of polar cod found in echo-sounder transects and pelagic trawls throughout the Barents Sea[17], identical to the method used for estimating capelin (*Mallotus villosus*) stock size in the Barents Sea[40].

**Ocean circulation model and ice module**. The hydrodynamic model used to represent the currents and oceanographic conditions (i.e. temperature, salinity, and ice concentration/cover) in the study area was based on the Regional Ocean Modeling System (ROMS, http://myroms.org), a free-surface, hydrostatic, primitive equation ocean general circulation model[41,42]. The ROMS model was run with a horizontal resolution of 4 × 4 km in an orthogonal, curvilinear grid covering parts of the North Atlantic and all the Nordic and Barents seas (see inset in Fig. 1 for extent of ROMS model) over the time period 1960–2017[20,43–45]. The output from ROMS contained velocity fields, ice concentration, temperature, and salinity in 32 terrain following vertical layers, and a temporal resolution of 24 h.

**Drift simulations and search algorithm**. The advection of particles in the horizontal plane was modelled by the Runge-Kutta fourth order scheme LADIM[46,47]. As early life stages of polar cod are usually found close to the surface[13], particles were uniformly distributed in the upper 10 meters with a fixed depth throughout the drift phase from 1 January to 30 September. In an exhaustive search for potential spawning areas of polar cod in the Barents Sea, particles were released in a regular grid (≈40 km equidistance, 537 positions in total) across the entire Barents Sea shelf shallower than 400 m that had been covered by an ice concentration of more than 15% in the period 1990–2017 (see extent of release grid in Fig. 2). A new ensemble of 100 particles were released at every point in the grid, every day from 1 January to 30 April, repeated for every year between 1990 and 2017 (yielding a total of 639,030 particles each year). Subsequently, an objective search algorithm identified drift trajectories that intersected the 0-group observations of the autumn survey within a three-week period of the surveys. The ability of the drift trajectories to explain the observed 0-group abundance and distribution was thus interpreted as a confirmation of spawning at a given release point and a high larval survival integrated over the drift phase. To allow a direct comparison between number of simulated drift intersections and 0-group abundance, both indices were log-transformed and scaled between 0 and 1. In line with the hypothesis of ice as a prerequisite for spawning, the drift trajectories' ability to predict the observed 0-group abundance was weighted by ice concentration at drift start point (i.e. at spawning area). To elucidate on the possible effects of heating on recruitment we extracted temperature profiles from all individual drift trajectories, and to decrease the effect of minor cold spells or heat waves on the subsequent analysis we applied a 10-day moving average filter on the temperature profiles.

**Statistical modelling**. A probabilistic map of 0-group polar cod presence was calculated by using a two-dimensional binomial GAM smoother, based on the geographical coordinates of pelagic trawls, presence-absence of polar cod larvae in the pelagic trawls, and using the logit-link function as implemented in R-package "mgvc"[48]. Moreover, to quantify the effect of environmental conditions on larval survival/recruitment strength, we fitted a linear regression model with recruitment strength as independent variable with the covariates Barents Sea ice cover (area of the Barents Sea covered by ice concentration higher than 15%, extracted from the ROMS model), maximum temperature encountered by larvae (10-day mean-filtered over 100 larvae released from the most likely spawning area for a given year), and estimated TSB. This regression model was fitted separately for the north-western (Svalbard) and south-eastern (Pechora Sea) spawning areas as implemented in the base R-package "stats"[49]. In the model selection phase, we applied a stepwise model selection scheme with the initial inclusion of all relevant variables, where only the variables deemed significant was included in the final model. Due to the high degree of collinearity

between some of the variables, we also did a variance partitioning analysis to disentangle the separate and/or common effects of the variables[50].

**Reporting summary**. Further information on research design is available in the Nature Research Reporting Summary linked to this article.

## Data availability
The datasets generated during and/or analysed during the current study are not publicly available due to large file sizes but are available from the corresponding author on reasonable request.

## Code availability
Custom code and algorithms used for analyses in this study will be made available on request to the authors.

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

## Author contributions

M.B.O.H., E.E., H.G., and F.V. conceived the study, interpreted the data and wrote the paper. M.B.O.H designed and performed the model simulations and statistical analyses.

## Competing interests

The authors declare no competing interests.
