## [Peer Review File · Communications Biology]

Reviewers' comments:

Reviewer #1 (Remarks to the Author):

This study explores the polar cod variation in relation to Arctic sea ice condition. The topic is attractive, but the manuscript is not well written and presented. The arguments are not solid and sometimes are speculated. The results presented seem to diverge from the topic and aims. Detailed comments are given below:

Comments:

1. There are some idiosyncrasies in English, which occasionally make manuscript difficult to read.
2. Page 1 describes the idea clearly, but pity that the following few pages seem to diverse and sometimes confusing.
3. The title and the aim of this study propose the importance of global warming and retreating ice, yet the results focus on the different scenario of interannual variation. Title and the aims in the first page should be modified in order to fit the main findings.
4. L38, the meaning of "true" Arctic species is not clear.
5. L41-42, as this sentence is for supporting the argument that cod is adapted to ice, it may be better to modify to: low mortality in lower temperature. By the way, warmer than typical of Arctic water masses seems to infer water masses from outside of Arctic? I guess the warming of local Arctic water is also possible to increase the mortality?
6. L43-45, does this mean the spawning had been observed in the past?
7. L52, because only 2 data are provided, this seems cod is increasing in time?
8. It will be good to indicate the two inflows that influence ice cover of the Barents Sea, as well as the location of Polar Fronts in Fig.1.
9. L68-69, Fig. 1 did not show the variation of ice as indicated in the text. Maybe can show the averaged, maximum and minimum ice extension in fig 1.
10. Backward tracking is used to deduce the potential spawning area. Apart from ocean currents, does particle movement depends on ice concentration, as the early section describes that eggs tend to spawn under the ice?
11. Fig. 3, what are green and blue circles represent, respectively? Gray trajectories are difficult to see, as the ice cover is also shaded in gray. Please also adjust the color scale of temperature, most of the area is red, so cannot see the difference clearly. The location and major currents mentioned in the manuscript should also be marked in the figure.
12. L130, the decadal trend cannot be seen from fig. 4. Time series may be a better option to show the trend.
13. Fig. 4, numbers are not clear, and the size of the circle should be more distinct and also provide a scale. Blue shading (single color shading) is difficult to see the variance.
14. What is Atlantic-like dynamic? And what is Atlantic-like water?

Reviewer #2 (Remarks to the Author):

Arctic amplification of global warming is leading to a reduction of sea-ice extent and duration. This is apparent in the Barents Sea and may have consequences for early life stages of Polar cod which are likely ice-obligate and also central to the Arctic food web. The authors examine the inter-annual variability in ice cover with the variability in recruitment, SSB and spawning distribution. They use a hydrodynamic model and particle trajectory simulations to back-track potential spawning locations based on 0-group polar cod observed in the Autumn surveys. Two spawning regions have been

identified in the Barents Sea: one in the northwest, east of Svalbard, and one in the southeast in the Pechora Sea. There was no observed correlation with ice cover and recruitment east of Svalbard. However, ice consistently forms there. They did find a correlation with annual ice cover and the magnitude of spawning activity and larval survival in the Pechora Sea. The authors suggest that the recruitment variability is driven by a match/mismatch with copepod nauplii with mismatches likely occurring in years with reduced ice cover. They are unclear if this is caused by reduction in ice-cover or an increase in temperature, as the two variables are collinear. The authors conclude as the climate continues to warm the spawning region near Svalbard will likely continue to persist, while the Pechora Sea may become unsuitable.

Overall, this study examines a key species of the Arctic marine ecosystems and its response to climate warming. The authors look at potential mechanisms that may be driving Polar cod's larval survival, spawning activity and recruitment and how they differ in two spawning regions. They take care to explain the importance linking physical changes to the biological responses and how they vary by region. Overall, the manuscript is well thought out and written, and would be a good contribution to Communications Biology. The manuscript can be improved by adding more details to the methods section (see specific comments below) for reproducibility. More clearly defining the specific objectives and statistical tests would also be helpful.

I have a few suggestions for the manuscript:

1. Authors switch back and forth between 0-gr and 0-group, just stick with one.
2. Line 28, it would be helpful to include general timeframe for the "inception of satellite monitoring".
3. Likewise, what do the authors mean by the "majority of reduction and inter-annual variability observed in the Barents Sea." It would be helpful if they'd be more specific.
4. It would be helpful if the authors included more details in the methods. For example, GAM equations, equations for estimating SSB, more about the correlation analysis and the lags.
5. In the methods, they mention that the GAM model explained 40% of the deviance. That belongs in the results
6. Also to be clear, the same starting spawning grid was used for every year? If so, make this more clear (maybe include the grid as a supplemental figure or even state the number of grid points).
7. Where does the temperature data come from?
8. In figure 2 the line for SSB looks pink, not purple.
9. Figure 3 almost has too much information that it's a little confusing, especially the grey circles, gradient of grey and grey lines. It's impressive how much information is packed in that paneled figure.
10. In the discussion (or introduction or methods), authors should mention the prior work by Eriksen et al. (2015) that looked into the correlation of ice-cover and age-0 polar cod in the Barents Sea. The current study takes a much deeper look into this, but it would be worth mentioning the findings of prior study.
11. The article formatting needs to adhere to Communications Biology rules, such as, no citations in the abstract, etc...

Referee expertise:

Referee #1: particle tracking ocean circulation models

Referee #2: arctic cod, general ecology

Reviewers' comments:

Reviewer #1 (Remarks to the Author):

This study explores the polar cod variation in relation to Arctic sea ice condition. The topic is attractive, but the manuscript is not well written and presented. The arguments are not solid and sometimes are speculated. The results presented seem to diverge from the topic and aims. Detailed comments are given below:

Comments:

1. There are some idiosyncrasies in English, which occasionally make manuscript difficult to read.

Author's reply: After a thorough revision with this in mind we believe that the idiosyncrasies have been resolved making the manuscript easier to read.

2. Page 1 describes the idea clearly, but pity that the following few pages seem to diverse and sometimes confusing.

Author's reply: Through careful revision we are confident that the results and discussion align more with the aims presented in the introductory paragraphs, and that our phrasing and presentation of the results are more concise.

3. The title and the aim of this study propose the importance of global warming and retreating ice, yet the results focus on the different scenario of interannual variation. Title and the aims in the first page should be modified in order to fit the main findings.

Author's reply: We tend to disagree with this statement. Since a firm relationship was found between ice/heat and recruitment and plausible mechanistic explanations back up our findings, we believe that it is meaningful to put this in the context of climate change and ice retreat. Although the declining trend in yearly maximum ice cover in the study period was admittedly weak, yet significant (see L136-L138), we still feel warranted to state this as strongly as we have done. However, to introduce the idea of interannual variation earlier we now also mention this in the abstract: "... Here we show how **variability and** change in sea ice cover in the Barents Sea, the largest shelf sea of the Arctic, affect the population

dynamics of a keystone species of the ice-associated food web, the polar cod ..." (L10-L13)

4. L38, the meaning of "true" Arctic species is not clear.

Author's reply: We agree that this phrase is imprecise, it now reads "The polar cod is endemic to the Arctic ..." (L31-L32)

5. L41-42, as this sentence is for supporting the argument that cod is adapted to ice, it may be better to modify to: low mortality in lower temperature. By the way, warmer than typical of Arctic water masses seems to infer water masses from outside of Arctic? I guess the warming of local Arctic water is also possible to increase the mortality?

Author's reply: We agree that this could be phrased better; now revised to: "... and low mortality of larvae in the close to freezing temperatures typical of Arctic water masses ..." (L35-L36). Moreover, although the primary driver of variation in ice-cover and temperature in the Barents Sea appears to be through advection of heat with the Atlantic water; intuitively, local heating of the water masses occupied by larvae in late summer, e.g. in the Novaya Zemlya Current or in the shallow Pechora Sea, may also affect their survival. At the same time we found no published studies related to local heating in the area. Thus, we changed our phrasing to a more general statement without implying water-mass or processes related to the high summer temperatures modelled in some years (L171-L173).

6. L43-45, does this mean the spawning had been observed in the past?

Author's reply: There have been no peer reviewed articles that have directly reported spawning west of Svalbard, thus trying to locate the spawning areas was a sub-goal of the article (as implied between L39-L43, and stated between L65-L67). However, as pointed out between L39 and L43, Ponomarenko (1968) reviewed the Soviet literature and compiled the anecdotes/knowledge accumulated by Soviet scientists stationed at various research stations along the Russian Arctic coast.

7. L52, because only 2 data are provided, this seems cod is increasing in time?

Author's reply: Yes, two data points was provided here, as examples of the extreme variation. To clarify misunderstandings regarding a potential increase we added at the end of the sentence: "... yet with no clear trend" (L46-L47).

8. It will be good to indicate the two inflows that influence ice cover of the Barents Sea, as well as the location of Polar Fronts in Fig.1.

Author's reply: Duly noted, to give the reader an overview of the most important oceanographic features of the study area we made a new figure 1, presenting: arrows indicating currents, lines indicating the Polar Front,

and colored gradient representing variation in ice concentration. All information contained in the old figure 1 has now been moved to figure 2.

9. L68-69, Fig. 1 did not show the variation of ice as indicated in the text. Maybe can show the averaged, maximum and minimum ice extension in fig 1.

Author's reply: See previous comment.

10. Backward tracking is used to deduce the potential spawning area. Apart from ocean currents, does particle movement depends on ice concentration, as the early section describes that eggs tend to spawn under the ice?

Author's reply: At present there are no published literature on the specific gravity (i.e. density) of polar cod eggs, nor on particular vertical movement/behavior of larvae. We thus took a conservative approach regarding vertical position of propagules and used fixed depths from 0-10m, as larvae are reported to stay near the surface and the juveniles are usually found in the upper 20m in our ecosystem survey. Thus, in short, eggs are not influenced by ice directly in the model. However, within the ROMS model we expect less wind driven currents underneath the ice generally, leading to reduced advection underneath the ice compared to open water.

11. Fig. 3, what are green and blue circles represent, respectively? Gray trajectories are difficult to see, as the ice cover is also shaded in gray. Please also adjust the color scale of temperature, most of the area is red, so cannot see the difference clearly. The location and major currents mentioned in the manuscript should also be marked in the figure.

Author's reply: Figure 3 has been revised, and hopefully there are now no confusions or unclarities regarding layout. Since arrows indicating general circulation is now included in figure 1, we feel that no further cluttering of figure 3 is necessary.

12. L130, the decadal trend cannot be seen from fig. 4. Time series may be a better option to show the trend.

Author's reply: Yes, we agree. Figure 4 has now been revised to depict the northward trend more clearly, along with more text describing the trend (L122-L125).

13. Fig. 4, numbers are not clear, and the size of the circle should be more distinct and also provide a scale. Blue shading (single color shading) is difficult to see the variance.

Author's reply: This figure has been revised, see comment above. We believe the figure now purvey the message more clearly.

14. What is Atlantic-like dynamic? And what is Atlantic-like water?

Author's reply: We are sorry for the imprecise choice of words. What we meant by Atlantic-like dynamic was Atlantic-like *bloom* dynamics of the phytoplankton (see changes in text, L162-L164). Also, what we meant by Atlantic-like water was *warmer* water. This has been resolved wherever this dubious word appeared.

Reviewer #2 (Remarks to the Author):

Arctic amplification of global warming is leading to a reduction of sea-ice extent and duration. This is apparent in the Barents Sea and may have consequences for early life stages of Polar cod which are likely ice-obligate and also central to the Arctic food web. The authors examine the inter-annual variability in ice cover with the variability in recruitment, SSB and spawning distribution. They use a hydrodynamic model and particle trajectory simulations to back-track potential spawning locations based on 0-group polar cod observed in the Autumn surveys. Two spawning regions have been identified in the Barents Sea: one in the northwest, east of Svalbard, and one in the southeast in the Pechora Sea. There was no observed correlation with ice cover and recruitment east of Svalbard. However, ice consistently forms there. They did find a correlation with annual ice cover and the magnitude of spawning activity and larval survival in the Pechora Sea. The authors suggest that the recruitment variability is driven by a match/mismatch with copepod nauplii with mismatches likely occurring in years with reduced ice cover. They are unclear if this is caused by reduction in ice-cover or an increase in temperature, as the two variables are collinear. The authors conclude as the climate continues to warm the spawning region near Svalbard will likely continue to persist, while the Pechora Sea may become unsuitable.

Overall, this study examines a key species of the Arctic marine ecosystems and its response to climate warming. The authors look at potential mechanisms that may be driving Polar cod's larval survival, spawning activity and recruitment and how they differ in two spawning regions. They take care to explain the importance linking physical changes to the biological responses and how they vary by region. Overall, the manuscript is well thought out and written, and would be a good contribution to Communications Biology. The manuscript can be improved by adding more details to the methods section (see specific comments below) for reproducibility. More clearly defining the specific objectives and statistical tests would also be helpful.

Author's reply: Thank you for your constructive summary and comments. The methods and statistical objectives/tests have now been more clearly defined in the methods, and made the study more easily reproducible.

I have a few suggestions for the manuscript:

1. Authors switch back and forth between 0-gr and 0-group, just stick with one.

Author's reply: Agree, we stick with 0-group.

2. Line 28, it would be helpful to include general timeframe for the “inception of satellite monitoring”.

Author's reply: Agree, the introductory statement has been revised, and a time of reference has now been included (see L23-24).

3. Likewise, what do the authors mean by the “majority of reduction and inter-annual variability observed in the Barents Sea.” It would be helpful if they'd be more specific.

Author's reply: As comment above, we agree that the opening statement in the introduction was not good enough, and the revised sentence now reads: “The Arctic Ocean winter sea ice cover has steadily declined since the 1970s, and the majority of this reduction has been observed in the Barents Sea” (see L23-24).

4. It would be helpful if the authors included more details in the methods. For example, GAM equations, equations for estimating SSB, more about the correlation analysis and the lags.

Author's reply: More information on the methods has now been included, and the methods should now be complete. However, as the equations used to estimate TSB was not original research performed in this study, we believe that it is more appropriate to refer to Gjøsæter et al. (2002) that reported on the methods the first time, and we also refer to a book chapter regarding the survey, where the total stock biomass was taken from. Also, to clarify the GAM equations, the variables and specifications used in the GAM are now written out in the methods (L267-L270).

5. In the methods, they mention that the GAM model explained 40% of the deviance. That belongs in the results

Author's reply: Agree, the GAM results have been moved to the results section.

6. Also to be clear, the same starting spawning grid was used for every year? If so, make this more clear (maybe include the grid as a supplemental figure or even state the number of grid points).

Author's reply: The number of release points is now stated in methods (L246), and the release grid can now be seen in figure 2, first referred to and introduced in the final section of the introduction (L69-74).

7. Where does the temperature data come from?

Author's reply: All temperature data comes from the ROMS ocean model, which has been shown to replicate the observed variability in temperature to a high degree (see L231-L238 for details on ROMS model). The information on data sources is now briefly mentioned in the last paragraph of the introduction (L67-L69), and in more detail in the methods section specific to the ROMS model (L231-L238).

8. In figure 2 the line for SSB looks pink, not purple.

Author's reply: The line is now described as pink

9. Figure 3 almost has too much information that it's a little confusing, especially the grey circles, gradient of grey and grey lines. It's impressive how much information is packed in that paneled figure.

Author's reply: Yes, the figure may contain too much information, and it has now been revised for clarity. We believe the figure is now more clear.

10. In the discussion (or introduction or methods), authors should mention the prior work by Eriksen et al. (2015) that looked into the correlation of ice-cover and age-0 polar cod in the Barents Sea. The current study takes a much deeper look into this, but it would be worth mentioning the findings of prior study.

Author's reply: We agree, Eriksen et al. (2015) is now mentioned when introducing the relationship between ice and recruitment (see L52).

11. The article formatting needs to adhere to Communications Biology rules, such as, no citations in the abstract, etc...

Author's reply: During the revision process the manuscript now adhere to COMMS BIOL formatting standards.

REVIEWERS' COMMENTS:

Reviewer #1 (Remarks to the Author):

The revision had generally addressed my previous questions. A few minor modifications are required before acceptance.

Minor comments:

1. L33&L37: these two sentences are confusing: "from spawning of eggs under ice" & "no direct observation of spawning". Is "the spawning of eggs under ice" the result of farming or laboratory experiment?
2. L62, instead of space, please use comma instead, i.e. 632,304 km²
3. L69, since ice concentration, temperature, and salinity are not used for particle tracking, please remove those terms listed here to avoid confusion.
4. L129, L132, L138, L140, L142, L144, and throughout the manuscript...:abbreviations used here should be defined and also unified, or provide a table, i.e. what is t, and df? and "r" may be more common than "cor" for representing correlation coefficient; unify p-value and p...
5. L176, listed the 5 lowest recruitment years
6. L193, westerly -> westward
7. Fig.4, please mark the year number on the dots.
8. L233, add "based on" before Regional Ocean Model System
9. L237, add the longitude and latitude range for indicating model domain, and also add the time period covered by the model.

Reviewer #2 (Remarks to the Author):

The authors did an excellent job of addressing comments, especially regarding the figures. The additional figure makes the paper a lot clearer and more informative. The added details in the methods section help with reproducibility. Reading through, the revised article, I have a few minor comments and suggestions, please see below.

1. Line 21 perhaps change development to propagation.
2. Line 43 What do the authors mean by vital rates? Mortality? Growth?
3. Line 65 I am not sure if I agree with the phrasing "locate the hitherto hidden spawning locations of polar cod under the ice." The authors are using models to provide plausible estimates of spawning locations, not directly sampling the spawning fish. Further, this discounts prior observations/anecdotes mentioned in Ponomarenko (1968) which the authors cite. Consider changing the wording.
4. In the statistical modeling section of the methods, the authors should mention which statistical software used for analysis. If R, cite the packages.

5. Lines 280-281 Add a citation for variance partitioning analysis.
6. Fig 3 overlap of latitude and longitude labels at 81 degrees N and 20 degrees.

REVIEWERS' COMMENTS:

Reviewer #1 (Remarks to the Author):

The revision had generally addressed my previous questions. A few minor modifications are required before acceptance.

Minor comments:

1. L33&L37: these two sentences are confusing: "from spawning of eggs under ice" & "no direct observation of spawning". Is "the spawning of eggs under ice" the result of farming or laboratory experiment?

Reply: Ponomarenko (1968) summarize personal observations and data contained in 150 reports from Soviet research stations across Soviet Arctic from period 1959-1965, and states that polar cod spawning usually takes place under the ice, close to the ice edge, or in areas where ice forms later in the season. Later we state that no observations of spawning have been done in modern times, and reiterate that the Ponomarenko paper from 1968 remains the single most complete description of polar cod stock structure and spawning fidelity. This confusion was also brought up by reviewer #2—thus to clarify we removed the confusing sentence that spawning has not been observed, it is now only implied that observations are scarce and not available since the early 1960s (L37-L41).

2. L62, instead of space, please use comma instead, i.e. 632,304 km²

Reply: Duly noted, thousands are now delimited by commas.

3. L69, since ice concentration, temperature, and salinity are not used for particle tracking, please remove those terms listed here to avoid confusion.

Reply: We agree, these variables are now only mentioned in the methods when describing the model.

4. L129, L132, L138, L140, L142, L144, and throughout the manuscript...:abbreviations used here should be defined and also unified, or provide a table, i.e. what is t, and df? and "r" may be more common than "cor" for representing correlation coefficient; unify p-value and p...

Reply: Abbreviations should now be unified. However we feel that terminology when reporting results from simple correlation tests and linear regression models should be a bare minimum of common statistical knowledge (e.g. r=correlation coefficient, t=t-value from t-test, df=degrees of freedom, p=p-value, F=F-value from ANOVA) and is made especially clear when put in context with previous sentence (i.e. a t-value associated with a correlation test or a significance test, F-value reported from a regression model etc.).

5. L176, listed the 5 lowest recruitment years

Reply: Yes this is correct, five of the years of lowest recruitment was found among the seven warmest summers—thus not a 1:1 relationship but a clear indication of the effect of warm summer temperatures on recruitment.

6. L193, westerly -> westward

Reply: To avoid confusion westward is changed with westerly

7. Fig.4, please mark the year number on the dots.

Reply: The dots have now been numbered by year

8. L233, add “based on” before Regional Ocean Model System

Reply: “based on” has now added before the description of the Regional Ocean Model System

9. L237, add the longitude and latitude range for indicating model domain, and also add the time period covered by the model.

Reply: Since the boundaries of an orthogonal, curvilinear grid is hard to describe with longitude and latitude, the area covered by the model is now indicated in small inset in Fig. 1.

Reviewer #2 (Remarks to the Author):

The authors did an excellent job of addressing comments, especially regarding the figures. The additional figure makes the paper a lot clearer and more informative. The added details in the methods section help with reproducibility. Reading through, the revised article, I have a few minor comments and suggestions, please see below.

1. Line 21 perhaps change development to propagation.

Reply: Agree, this has now been changed

2. Line 43 What do the authors mean by vital rates? Mortality? Growth?

Reply: Vital rates has been removed, and the sentence now reads: “The Barents Sea polar cod stock have been monitored annually by a joint Norwegian-Russian survey since 1986 ...” (L42-L43)

3. Line 65 I am not sure if I agree with the phrasing “locate the hitherto hidden spawning locations of polar cod under the ice.” The authors are using models to provide plausible estimates of spawning locations, not directly sampling the spawning fish. Further, this discounts prior observations/anecdotes mentioned in Ponomarenko (1968) which the authors cite. Consider changing the wording.

Reply: To avoid discounting previous arguments regarding Ponomarenko (1968) the sentence now plainly states what we did: “To locate the spawning locations of polar cod under the ice we “back-tracked” larval drift trajectories from observed 0-group polar cod in autumn to the most probable spawning locations in spring” (L75-L77)

4. In the statistical modeling section of the methods, the authors should mention which statistical software used for analysis. If R, cite the packages.

Reply: Various R packages are now cited where used.

5. Lines 280-281 Add a citation for variance partitioning analysis.

Reply: A reference to the principle of variance partitioning has now been added.

6. Fig 3 overlap of latitude and longitude labels at 81 degrees N and 20 degrees.

Reply: Figure 3 has now been revised and these labels do not overlap anymore.